# Venovenous bypass in adult liver transplant recipients: A single-center observational case series

Laurence Weinberg[1,2]*, Rebecca Caragata[1], Riley Hazard[1], Jarryd Ludski[1], Dong-Kyu Lee[3], Hugh Slifirski[1], Patrick Nugraha[1], Daniel Do[1], Wendell Zhang[1], Robert Nicolae[1], Peter Kaldas[4], Michael A. Fink[4], Marcos V. Perini[4]

1 Department of Anaesthesia, Austin Health, Heidelberg, Australia, 2 Department of Critical Care, Austin Health, The University of Melbourne, Heidelberg, Australia, 3 Department of Anesthesiology and Pain Medicine, Dongguk University Ilsan Hospital, Goyang, Republic of Korea, 4 Department of Surgery, Austin Health, The University of Melbourne, Heidelberg, Australia

* laurence.weinberg@austin.org.au

## Abstract

### Background

Very little information is currently available on the use and outcomes of venovenous bypass (VVB) in liver transplantation (LT) in adults in Australia. In this study, we explored the indications, intraoperative course, and postoperative outcomes of patients who underwent VVB in a high-volume LT unit.

### Methods

The study was a single-center, retrospective observational case series of adult patients who underwent VVB during LT at Austin Health in Melbourne, Australia between March 2008 and March 2022. Information on baseline preoperative status and intraoperative variables, including specific VVB characteristics as well as postoperative and VVB-related complications was collected. The lengths of intensive care unit and hospital stays as well as intraoperative and in-hospital mortality were recorded.

### Results

Of the 900 LTs performed at this center during the aforementioned 14-year period, 27 (3%) included a VVB procedure. VVB was performed electively in 16 of these 27 patients (59.3%) and as a rescue technique to control massive bleeding in the other 11 (40.1%). The median (interquartile range [IQR]) age of those who underwent VVB procedures was 48 (39–55) years; the median age was 56 (47–62) years in the non-VVB group ($p$<0.0001). The median model for end-stage liver disease (MELD) scores were similar between the two patient groups. Complete blood data was available for 622 non-VVB patients. Twenty-six VVB (96.3%) and 603 non-VVB (96.9%) patients required intraoperative blood transfusions. The median (IQR) number of units of packed red blood cells transfused was 7 (4.8–12.5) units in the VVB group compared to 3.0 units (1.0–6.0) in the non-VVB group ($p$<0.0001). Inpatient

**Data Availability Statement:** All relevant data are within the manuscript and its Supporting information files.

**Funding:** The authors received no specific funding for this work.

**Competing interests:** The authors have declared that no competing interests exist.

mortality was 18.5% and 1.1% for the VVB and non-VVB groups, respectively (*p*<0.0001). There were no significant differences in length of hospital stay or incidence of acute kidney injury, primary graft dysfunction, or long-term graft failure between the two groups. Patients in the VVB group experienced a higher rate of postoperative non-anastomotic biliary stricture compared to patients in the non-VVB group (33% and 7.9%, respectively; *p* = 0.0003).

## Conclusions

VVB continues to play a vital role in LT. This case series highlights the heightened risk of major complications linked to VVB. However, the global transition to selective use of VVB underscores the urgent need for collaborative multi-center studies designed to address outstanding questions and parameters related to the safe implementation of this procedure.

## Introduction

Traditional surgical techniques used for liver transplantation (LT) procedures involve full cross-clamping and resection of the retrohepatic inferior vena cava (IVC) and temporary clamping of the portal vein. This can result in significant hemodynamic perturbations, including a reduction in venous return and cardiac output, distal venous congestion, and organ hypoperfusion [1]. Veno-venous bypass (VVB) was first popularized in the 1980s in an attempt to minimize these physiologic insults and maintain intraoperative hemodynamic stability [2]. Many variations in VVB set-up and technique as well as numerous approaches to vascular access have been developed [3, 4].

The performance of caval anastomoses with or without VVB continues to be a subject of debate. A rigorous Cochrane analysis examined the advantages and disadvantages of this procedure and reported that there was insufficient evidence available to support or dismiss the use of VVB in LT [5]. As part of this analysis, the authors evaluated the evidence that directly compared LT procedures performed with or without VVB. Of note, no significant differences in renal failure or blood transfusion requirements were identified. However, all trials identified featured a high risk of bias, and none reported on patient or graft survival.

More recently, an expert panel at a consensus conference on Enhanced Recovery for Liver Transplantation evaluated the routine application of VVB during LT and identified length of hospital stay, duration of surgery, units of packed red blood cells (PRBCs) transfused, early complication rates and mortality, and renal dysfunction as critical significant outcomes for patients undergoing VVB [6]. A comprehensive re-evaluation of the current literature and assessments focused on VVB during LT revealed that the quality of evidence for all outcomes of significance was extremely poor. The authors reported that most of the studies that compared outcomes of LT procedures involving VVB to those with no VVB reported no significant differences in duration of surgery [7–13], units of PRBCs transfused [7–14], early postoperative mortality [9, 12, 14], postoperative acute kidney injury [7–9, 11–13, 15, 16], early complication rates [10, 12, 16], or length of hospital stay [8, 11, 12]. The expert panel advised against the routine implementation of VVB, but acknowledged that there are specific circumstances in which its use may be justified.

The potential benefits of VVB in a select patient cohort include improved hemodynamic stability during the anhepatic phase secondary to preservation of venous return, decreased blood loss associated with portal hypertension, and preservation of blood flow to other vital

organs [17, 18]. Furthermore, some patients with metabolic or cholestatic liver disease, even in the absence of portal hypertension, are at an increased risk of portal congestion and may require VVB or a temporary portocaval shunt. VVB can promote diversion of blood flow in cases with complex anatomy and/or difficult surgical exposure (i.e., massive hepatomegaly), facilitate decompression of the portal-mesenteric circulation to reduce bacterial translocation secondary to intestinal congestion [19], and limit diffuse abdominal bleeding from venous collaterals [20]. Finally, VVB can also be considered in cases of fulminant hepatic failure as a means to maintain venous return, thereby reducing the need for excess fluids that can exacerbate intracranial hypertension and potential cerebral edema.

VVB has many associated disadvantages, with reported complication rates ranging from 10% to 30% [21]. Complications associated with vascular access include hematomas, wound infections, and lymphoceles as well as injuries to the brachial plexus and major vasculature. Additionally, the use of extracorporeal circuits may lead to hypothermia, air emboli, and thromboembolic complications. Overall, there is little to no high-quality evidence regarding the benefits and risks of VVB [6].

Advances in surgical methodology, notably, the increased use of caval-sparing surgical techniques (e.g., piggyback or cavo-cavostomy), had led to a gradual global decline in the routine use of VVB [22]. However, significant inter-institutional variations remain [23]. Recent surveys suggest that 42% of transplant centers across the United States and up to 38% of centers within European networks still offer selective use of VVB [24, 25].

At this time, there is little information regarding the uptake, safety, and outcomes of VVB in LT centers in Australia. This observational study was designed to characterize contemporary practices at a major Australian center and describe baseline characteristics, intraoperative course, specific VVB complications, and postoperative outcomes experienced by patients who have undergone LTs with VVB. As part of this case series, we aimed to explore ongoing indications for this procedure as well as the practicalities of VVB use in this modern era of selective use.

## Methods

### Ethical approval

This case series was registered retrospectively with the Australian New Zealand Clinical Trials Registry (no. 12623000499684) on May 16, 2023. Ethical approval was obtained from the Austin Health Human Research Ethics Committee on July 27, 2023 (HREC: 545–2022). The Ethics Committee waived the requirement for informed consent because the study was limited to the retrospective collection of de-identified data. There were no changes to the original study protocol at any stage of the work. Data analysis was performed only after obtaining ethical approval. Data were collected for research purposes on July 27, 2023. All data were de-identified and data analysis was completed on August 15, 2023.

### Study population, data sources, and variables

The study features a single-center retrospective, observational case series of patients who underwent a LT procedure at Austin Health between March 2008 and March 2022. Austin Health is a university teaching hospital located in Melbourne, Australia, and the site of the Victorian Liver Transplant Unit. More than 1,600 LTs have been performed to date at this unit. The unit currently performs ~100 LTs per year. The adult recipient program at this LT unit uses only deceased donor grafts. Inclusion criteria included adult patients (i.e., those ≥18 years of age) who received VVB as part of their LT procedure for any indication, including redo LTs, and patients who received elective (planned) or emergent (rescue) VVB. Patients

who did not receive VVB were included as controls. Paediatric LT recipients (i.e., those <18 years of age) were excluded.

Data were collected from the hospital's electronic medical record system and the prospectively maintained database of the transplant unit. Information was extracted regarding the prevalence of VVB, baseline patient characteristics and comorbidities, and preoperative biochemical, hematological, and coagulation profiles. Intraoperative variables included surgical time, indications for VVB use, nature of the bypass circuits, vascular access sites, laboratory findings, anesthesia management, administration of blood products, and fluid management. With respect to postoperative outcomes, we extracted information related to a broad range of perioperative complications, which were stratified by severity using the Clavien-Dindo classification system [26]. Finally, we evaluated the lengths of intensive care unit and hospital stays as well as intraoperative and in-hospital mortality. The study was conducted in accordance with Surgical CAse REport (SCARE) guidelines [27].

## VVB management

The decision to implement planned or emergency use of VVB for intraoperative rescue was made contemporaneously by the treating surgical and anesthesiology teams based on clinician preferences and clinical concerns. Planned VVB was considered in patients with severe portal hypertension in which a difficult hepatectomy was anticipated (i.e., total portal vein thrombosis with the need for vascular grafts or previous liver procedures in which decompression of the portal system had been performed to prevent excessive blood loss). Planned VVB was also considered for patients with massive livers, in which IVC clamping would be necessary for vascular occlusion needed to facilitate surgery (i.e., polycystic liver disease). In cases in which the diseased liver was so large that it would be difficult to access the major hepatic vasculature in the absence of portal hypertension, VVB was used to divert blood from the infrarenal IVC (and therefore the kidneys, lower limbs, and pelvis) to the right heart to facilitate decompression of the portal circulation and reduced congestion of the intestinal and splanchnic circulation.

Implementation of VVB was also considered in patients with pre-existing cardiac conditions, such as aortic stenosis or severe left ventricular outflow obstruction as a means to maintain adequate venous return to the right heart, thereby reducing the risk of cardiac decompensation and collapse with subsequent organ hypoperfusion. VVB was also employed as an "on-table" rescue intervention in patients who had undergone massive blood loss to facilitate vascular access needed to control bleeding while maintaining hemodynamic stability by ensuring adequate venous return to the right heart.

Likewise, the use of specific vascular access sites and techniques (cutdown *versus* percutaneous) were at the discretion of the treatment teams. When employed, the percutaneous approach was performed under ultrasound guidance; transesophageal echocardiography was used for the placement of all upper-body venous return lines. Cannulation was achieved using 16–18 Fr FemFlex cannulae (Edwards Lifesciences). A Cardiohelp™ (Maquet) extracorporeal membrane oxygenation (ECMO) console with an integrated centrifugal pump, oxygenator, and heat exchanger (HLS Module Advanced 7.0) was used at flow rates between 1–3 L/min. Heparin-bonded circuits were employed; systemic heparin was administered if required to target activated clotting time between 180 and 220 seconds. All VVB procedures were supervised by a dedicated clinical perfusionist.

## Statistical analysis

All statistical analysis was performed with GraphPad Prism version 10.2.2 for macOS (GraphPad Software, Boston, Massachusetts USA). Continuous variables were evaluated for normality

**Table 1. Baseline patient characteristics.** Data are presented as numbers (percentages) or medians (interquartile ranges) [minimum and maximum].

| Variable | VVB Group (n = 27) | Non-VVB Group (n = 873) | *P*-value |
|---|---|---|---|
| **Patient characteristics** | | | |
| Age (years) | 48 (39–55) [18–66] | 56 (47–62) [19–73] | <0.0001 |
| Body mass index (kg/m$^2$) | 25.0 (21.4–28.9) [18.2–34.1] | 27 (24–31) [16–50] | 0.001 |
| Smoking status within 1 year | 0 (0%) | 3 (0.3%) | >0.999 |
| **Causes of liver disease** | | | |
| Polycystic liver disease | 9 (33.3%) | 0 (0%) | <0.0001 |
| Primary biliary cirrhosis | 0 (0%) | 21 (2.3%) | >0.999 |
| Primary sclerosis cholangitis | 5 (18.5%) | 81 (9.0%) | 0.089 |
| Hepatitis B | 4 (14.8%) | 21 (2.3%) | 0.005 |
| Hepatitis C | 2 (7.4%) | 110 (12.3%) | 0.762 |
| Glycogen storage disease | 2 (7.4%) | 25 (2.8%) | 0.184 |
| Herpes simplex virus | 1 (3.7%) | 0 (0%) | 0.029 |
| Recurrent autoimmune hepatitis | 1 (3.7%) | 29 (3.2%) | 0.595 |
| Budd-Chiari syndrome | 1 (3.7%) | 1 (0.1%) | 0.058 |
| Non-alcoholic steatohepatitis | 1 (4%) | 10 (1.1%) | 0.279 |
| Cryptogenic cirrhosis | 0 (0%) | 26 (2.9%) | >0.999 |
| Malignancy | 0 (0%) | 189 (21.1%) | 0.003 |
| Acute hepatic necrosis | 0 (0%) | 51 (5.7%) | 0.393 |
| Alcohol | 1 (3.7%) | 113 (12.6%) | 0.240 |
| Other | 0 (0%) | 220 (24.5%) | 0.0009 |
| **Comorbidities** | | | |
| Model of end-stage liver disease (MELD) | 20 (9–26) [6–37] | 18 (13–24) [6–54] | 0.868 |
| American Society of Anesthesiologists Score 3 | 1 (3.7%) | 20 (2.2%) | 0.467 |
| American Society of Anesthesiologists Score 4 | 26 (96.3%) | 877 (97.7%) | 0.502 |
| **Laboratory results** | | | |
| Hemoglobin (g/L) | 94 (80–119) [65–141] | 98 (82–119) [54–177] | 0.652 |
| Platelet count (×10$^9$/L) | 115 (69–221) [18–396] | 74 (51–112) [14–940] | <0.0001 |
| Creatinine (μmol/L) | 81 (62–157) [43–718] | 92 (70–131) [31–948] | 0.569 |
| Estimated glomerular filtration rate (mL/min/1.73 m$^2$) | 79 (32–90) [5–90] | 79 (58–90) [13–90] | 0.193 |
| International normalized ratio (INR) | 1.4 (1.2–1.7) [1–4.8] | 1.6 (1.3–2.1) [0.9–8.7] | 0.039 |
| Bilirubin (μmol/L) | 23 (9–134) [5–819] | 97 (31–299) [4–1062] | <0.0001 |
| Albumin (g/L) | 31 (26–36) [16–41] | 29 (25–35) [15–50] | 0.454 |

assumption using Shapiro's test and visual check of Q-Q plots. Data are presented with mean ± standard deviation (SD), median (1st– 3rd quartiles) [minimun:maximum], or number of cases (percentile) for the descriptive statistics. Both the (single) t test (and nonparametric) analysis was used to compare two groups of values. Fishers exact test or the Chi-Square test was used to determine whether a statistically significant association existed between two categorical variables. A 2-sided P value below 0.05 was considered statistical significance based on the null hypothesis significance testing.

## Results

Of the 900 LT procedures performed at our center between March 2008 and March 2023, 27 (3%) included VVB. The information presented in Table 1 outlines the preoperative characteristics of these 900 patients. Patients who underwent LTs with VVB (VVB group) were younger and had lower body mass indexes (BMIs) than LT patients who did not undergo VVBs (non-

VVB group); the median (IQR) ages were 48 (39–55) years *versus* 56 years (47–62) years (p <0.0001), respectively, and the median BMIs were 25.0 (21.4–28.9) kg/m$^2$ *versus* 27 (24.1–31.2) kg/m$^2$ (*p* = 0.001). The underlying primary liver pathologies are also summarized in Table 1. The median (IQR) model for end-stage liver disease (MELD) scores were similar between the two groups, at 20 (9–26) *versus* 8 (13–24) for the VVB and non-VVB groups, respectively (*p* = 0.868). We observed no significant changes in the yearly median MELD scores of all patients in these groups over the entire 14-year period (*p* = 0.329).

Most of the VVB patients underwent caval-sparing surgical procedures (93%); only two patients (7%) were treated with conventional caval interposition. Table 2 summarizes the intraoperative course of this cohort, including the surgical and anesthetic management used in these cases. VVB was instituted as a planned procedure in 16 of 27 patients (59.3%); the remaining 11 (40.7%) required rescue VVB intraoperatively. Elective indications for VVB predominantly reflected anatomic complexity, including polycystic liver disease, massive hepatomegaly (from other causes), tumor proximity to the IVC (a large segment one lesion), and a previous transjugular intrahepatic portosystemic shunt procedure. In one case, VVB was initiated electively because of pre-existing cardiac disease (mixed aortic regurgitation and stenosis in a patient with a unicuspid valve). All 11 emergency cases underwent VVB to address massive intraoperative bleeding.

The different types of VVB circuits used at our center are shown in Figs 1 and 2. Most of the VVB procedures (85%) were performed with double-limb venous bypass circuits with concurrent systemic and portal venous drainage lines. Three patients were treated with single-limb circuits with either portal or systemic venous drainage alone (7% and 4%, respectively). Systemic venous drainage was achieved using an open surgical cut-down approach in 78% of the patients, most commonly via a groin incision targeting the femoral-saphenous veins or, in one case, via the infrahepatic IVC (within the operative field). Return venous cannulae were most commonly positioned in the internal jugular veins (63%) using a percutaneous approach. Surgical cut-downs of the left axillary vein were performed less frequently (33%). Detailed data related to the VVB circuit setup were missing from the handwritten operative report included in one patient record.

## Use of blood and blood products

The use of blood and blood products is summarized in Table 3. A similar proportion of patients received an intraoperative PRBC transfusion. However, the median (IQR) number of PRBCs transfused was 7 (4.8–12.5) units in the VVB group *versus* 3.0 (1.0–6.0) units in the non-VVB group (*p*<0.0001). While similar proportions of patients in each group received platelets and cryoprecipitate, a higher fraction of the patients in the VVB group (81.5%) received fresh frozen plasma compared to the non-VVB group (58%, *p* = 0.016). Of the patients who required blood or blood products, patients in the VVB group received significantly more units than those in the non-VVB group (Table 3). Intraoperative and postoperative arterial blood gas data are presented in the S1 File.

## Length of stay and complications

The perioperative complications and postoperative outcomes of patients in both the VVB and non-VVB groups are presented in Table 4. Two patients (7.4%) in the VVB group died during the procedure; both of these patients required emergency rescue VVB. Four other patients experienced complications that were directly attributed to the VVB procedure. One patient sustained a guidewire injury to the right heart that resulted in cardiac tamponade (Clavien Dindo Grade 3b), while another experienced an injury to the left subclavian artery that led to

**Table 2. Intraoperative variables of patients who underwent venovenous bypass (VVB).** Data are presented as numbers (proportions), medians (interquartile ranges), and [minimum and maximum].

| Variable | VVB group (n = 27) |
|---|---|
| Duration of surgery (minutes) | 660.0 (540.0–750.0) [360.0–1050.0] |
| Duration of VVB (minutes) | 124.0 (80.0–156.0) [30.0–330.0] |
| Lowest body temperature (ºC) | 35.5 (34.7–36.3) [31.8–37.0] |
| Highest body temperature (ºC) | 37.3 (36.8–38.0) [36.0–38.1] |
| **Surgical Technique** | |
| Caval-sparing (piggyback) | 25 (92.5%) |
| Caval interposition | 2 (7.4%) |
| **Elective *vs.* Emergency VVB** | |
| Elective (planned) | 16 (59.3%) |
| Emergent (rescue) | 11 (40.7%) |
| **Indication for VVB** (two patients exhibited more than one indication) | |
| Bleeding | 11 (40.7%) |
| Polycystic liver disease | 9 (33.3%) |
| Massive hepatomegaly (secondary to other causes) | 3 (11.1%) |
| Previous transjugular intrahepatic portosystemic shunt | 2 (7.4%) |
| Tumor proximity to the inferior vena cava (large segment one lesion) | 1 (3.7%) |
| Cardiac valvulopathy (mixed aortic regurgitation and stenosis) | 1 (3.7%) |
| Redo liver transplant | 2 (7.4%) |
| **VVB Circuit Type** | |
| Systemic and portal | 23 (85.2%) |
| Portal only | 2 (7.4%) |
| Systemic only | 1 (3.7%) |
| Missing data (insufficient description) | 1 (3.7%) |
| **Systemic Venous Drainage Access** | |
| Open cut-down | 21 (77.7%) |
| Percutaneous | 3 (11.1%) |
| Not applicable (portal drainage limb only) | 2 (7.4%) |
| Missing data (insufficient description) | 1 (3.7%) |
| **Venous Return Access** | |
| Percutaneous via internal jugular vein | 17 (63.0%) |
| Open cut-down via axillary vein | 9 (33.3%) |
| Missing data (insufficient description) | 1 (3.7%) |
| **Ionotropic support during VVB** | |
| **Norepinephrine** | |
| No. of patients | 26 (96.2%) |
| Dose (mg) | 8.6 (2.5–13.4) [0.8–24.0] |
| **Metaraminol** | |
| No. of patients | 9 (33.3%) |
| Dose (mg) | 320 (4.0–500.0) [1.5–1000.0] |
| **Adrenalin** | |
| No. of patients | 8 (29.6%) |
| Dose (mg) | 13.5 (10.0–26.3) [0.01–73] |
| **Vasopressin** | |
| No. of patients | 5 (18.5%) |
| Dose (units) | 1 (0.5–5.3) [0.2–9.0] |
| **Methylene blue** | |

*(Continued)*

**Table 2.** (Continued)

| Variable | VVB group (n = 27) |
| --- | --- |
| No. of patients | 2 (7.4%) |
| Dose (mg) | 75 (50–100) [50–100] |
| **Electrolyte replacement** | |
| **Sodium bicarbonate (8.4%)** | |
| No. of patients | 13 (48.1%) |
| Amount (mmol) | 126 (100–750) [30–2000] |
| **Calcium chloride** | |
| No. of patients | 18 (66.6%) |
| Amount (mmol) | 30 (20.75–40.25) [14–70] |
| **Anesthesia agents** | |
| **Inhaled anesthetic agent** | |
| No. of patients | 27 (100%) |
| Isoflurane | 6 (22.2%) |
| Sevoflurane | 21 (77.7%) |
| Fentanyl | |
| No. of patients | 27 (100%) |
| Dose (µg) | 500 (350.0–1000.0) [200.0–4500.0] |
| **Propofol** | |
| No. of patients | 18 (66.6%) |
| Dose (mg) | 100 (77.5–150.0) [24.0–250.0] |
| **Midazolam** | |
| No. of Patients | 15 (55.5%) |
| Dose (mg) | 5.0 (3.0–5.0) [1.0–20.0] |
| **Tranexamic acid** | |
| No. of Patients | 8 (29.6%) |
| Dose (g) | 1 (1.0–1.1) [1.0–2.0] |

hemothorax and significant blood loss (Clavien Dindo Grade 3b). Both complications occurred during the percutaneous insertion of the return cannula into the internal jugular veins. There was also a single case of a groin wound seroma related to a saphenous cut-down site, which did not require further management. One patient experienced a small-volume air embolism which resulted in transient intraoperative hypoxia and right heart dysfunction albeit with no long-term sequelae.

While a substantially higher proportion of patients in the VVB group needed to return to the operating room for treatment of postoperative bleeding (85.2% *versus* 13.6%, $p<0.0001$), the incidence of hepatic artery thrombosis, major neurological complications, acute kidney injury, and postoperative sepsis were similar between the two groups (Table 4). Although the inpatient mortality was 18.5% in the VVB group, compared to 1.1% in the non-VVB group ($p<0.0001$). we observed no significant differences in length of hospital stay, primary graft nonfunction, or long-term graft failure. Interestingly, patients in the VVB group had a higher rate of postoperative non-anastomotic biliary stricture compared to patients in the non-VVB group (33% *versus* 7.9%; $p = 0.0003$).

## Discussion

In this retrospective single-center study, 3% of patients undergoing LT experienced an intraoperative VVB procedure. While no significant differences were observed between the VVB

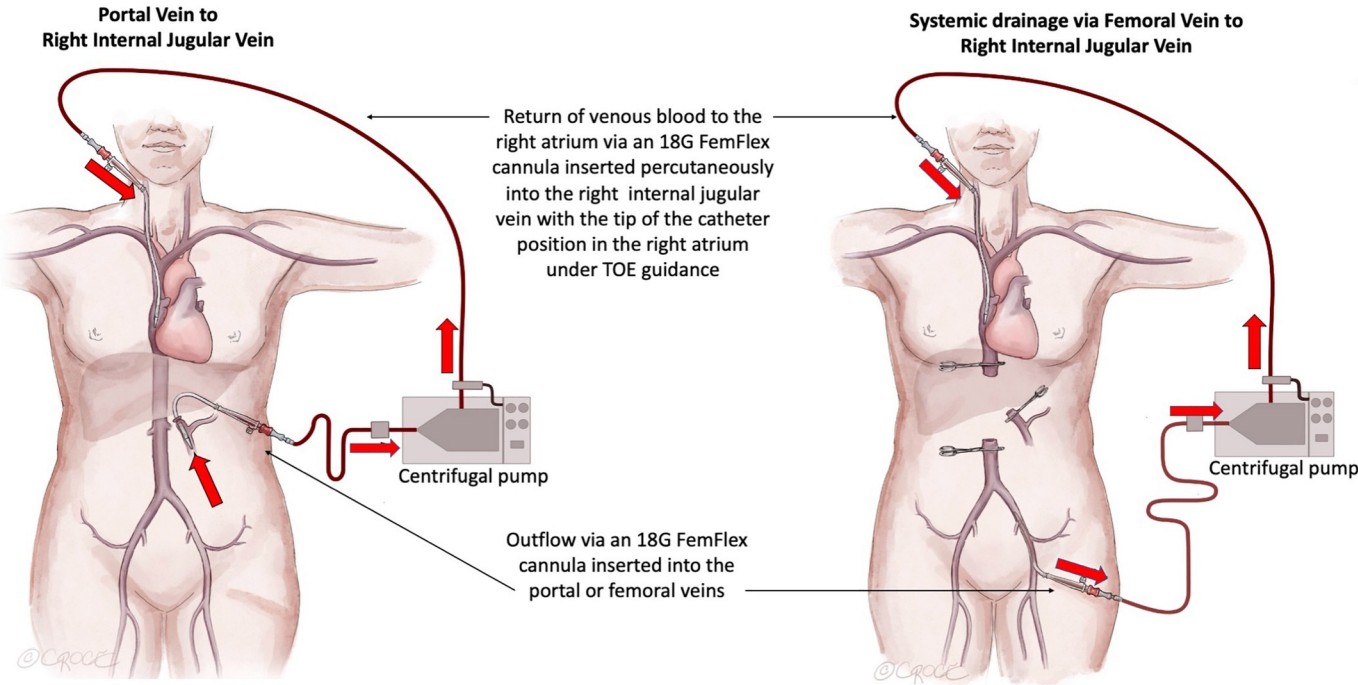

**Fig 1. Venovenous bypass circuits from the portal vein to the right internal jugular vein without inferior vena cava resection (left) and with systemic drainage via a femoral vein to the right internal jugular vein with an inferior vena cava resection (right).**

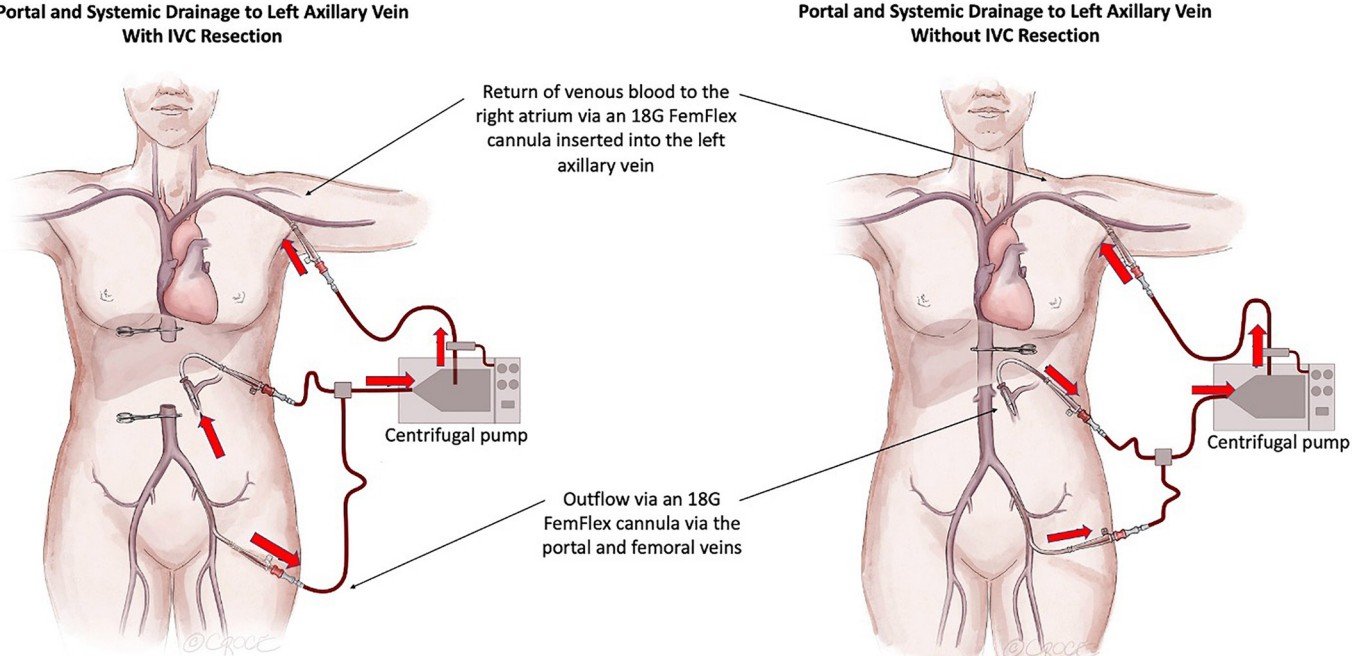

**Fig 2. Venovenous bypass circuit from the portal and femoral veins to the left axillary vein with and without an inferior vena cava resection.**

**Table 3. Intraoperative fluids, blood and blood products.** Only patients with complete medical records were included. Data are presented as number (proportion), median (interquartile ranges), and [minimum and maximum].

| | VVB Group (n = 27) | Non-VVB Group (n = 622) | *P*-value |
|---|---|---|---|
| **Intraoperative fluid** | | | |
| Total fluid including blood (excluding platelets, fresh frozen plasma, and cryoprecipitate) (L) | 9.9 (5.3–12.0) [2.3–63.0] | 7.4 (4.9–11.0) [1.0–53.4] | <0.0001 |
| **PlasmaLyte solution** | | | |
| No. of patients | 27 (100%) | 622 (100%) | >0.999 |
| Volume (L) | 6.0 (3.0–9.0) [1.0–46.0] | 4.0 (3.0–6.0) [0.4–125.0] | 0.077 |
| **Albumin 20%** | | | |
| No. of patients | 24 (88.9%) | 614 (98.7%) | 0.454 |
| Volume (L) | 1.3 (0.5–2.3) [0.1–8.0] | 0.8 (0.6–1.2) [0.2–5.0] | 0.057 |
| **Albumin 4%** | | | |
| No. of patients | 3 (11.1%) | 8 (1.3%) | 0.009 |
| Volume (L) | 2.0 (1.0–5.5) [1.0–5.5] | 1.5 (1.0–1.8) [1.0–2.0] | 0.356 |
| **Blood and blood products** | | | |
| **Packed red blood cells** | | | |
| No. of patients | 26 (96.3%) | 603 (96.9%) | 0.578 |
| Units | 7.0 (4.8–12.5) [2.0–30.0] | 3.0 (1.0–6.0) [1.0–31.0] | <0.0001 |
| **Fresh frozen plasma** | | | |
| No. of patients | 22 (81.5%) | 361 (58.0%) | 0.016 |
| Units | 4.0 (3.0–5.8) [2.0–9.0] | 3.0 (2.0–4.0) [0.0–12.0] | 0.003 |
| **Platelets** | | | |
| No. of patients | 18 (66.6%) | 344 (55.3%) | 0.323 |
| Units | 2.0 (2.0–4.0) [1.0–8.0] | 2.0 (1.0–2.0) [0.0–15.0] | 0.003 |
| **Cryoprecipitate** | | | |
| No. of patients | 16 (59.2%) | 300 (48.2%) | 0.326 |
| Units | 17.0 (7.5–24.8) [1.0–54.0] | 9.0 (5.0–10.0) [0–50] | 0.003 |
| **Autologous red cells via Cell Saver** | | | |
| No. of patients | 22 (81.4%) | 538 (86.5%) | 0.400 |
| Volume (L) | 4.5 (1.7–8.1) [0.5–17.3] | 1.2 (0.7–2.5) [0.0–14.9] | <0.0001 |
| **Donor-washed red cells** | | | |
| No. of patients | 14 (52%) | 267 (42.9%) | 0.429 |
| Volume (L) | 0.9 (0.5–0.9) [0.3–0.9] | 0.7 (0.5–0.9) [0.0–2.3] | 0.201 |

and non-VVB groups with respect to length of hospital stay, acute kidney injury, primary graft nonfunction, or long-term graft failure, VVB use was associated with a higher long-term incidence of non-anastomotic strictures and a greater need for blood products. The inpatient mortality of patients who underwent a VVB procedure was 18.5%. Our findings suggest that while VVB continues to play a vital role in LT in our center, the high rate of postoperative complications and perioperative mortality underscore an urgent need for collaborative multi-center studies that address appropriate implementation and patient selection.

There has been a global trend away from routine use of VVB, largely due to the absence of high-quality evidence supporting its benefit. An earlier Cochrane review [5] and a more recent systematic review [6] highlighted the limited evidence available that could be interpreted as supporting or opposing the routine use of VVB in LT, noting specifically the small number of published trials and the high risk of bias. Many of the technical aspects of LT procedures are based largely on individual institutional protocols and preferences, and recent survey data

**Table 4. Perioperative complications.** Data are presented as numbers (proportions) and medians (interquartile ranges [IQRs]).

| Complication type | VVB Group (n = 27) | Non-VVB Group (n = 622) | *P*-value |
|---|---|---|---|
| Return to theatre for bleeding | 23 (85.2%) | 85 (13.6%) | <0.0001 |
| Return to the operating room for infection | 0 (0.0%) | 21 (3.4%) | >0.999 |
| Hepatic artery or portal vein thrombosis | 2 (7.4%) | 26 (4.2%) | 0.327 |
| Liver abscess | 1 (3.7%) | 0 (0.0%) | 0.416 |
| Postoperative stroke | 1 (3.7%) | 2 (0.3%) | 0.112 |
| Prolonged respiratory failure requiring tracheostomy | 3 (11.1%) | 0 (0.0%) | <0.0001 |
| Acute kidney injury (Kidney Disease Improving Global Outcomes classification) | 5 (18.5%) | 134 (21.5%) | >0.999 |
| **Length of stay** | | | |
| Hospital stay (days) | 21 (15–31) [11–88] | 17 (12–330) [5–1172] | 0.057 |
| Intensive care unit stay (days) | 5 (3–11) [2–58] | 3.5 (2.0–6.60) [0.3–566] | 0.008 |
| **Graft outcomes** | | | |
| Non-anastomotic bile duct stricture | 9 (33.3) | 49 (7.9%) | 0.0003 |
| Primary graft nonfunction–retransplant within 30 days | 1 (3.7%) | 18 (2.9%) | 0.559 |
| Long-term graft failure | 1 (3.7%) | 15 (2.4%) | 0.497 |
| **Mortality** | | | |
| Death during the procedure | 2 (7.4%) | 2 (0.3%) | 0.0095 |
| Inpatient death | 5 (18.5%) | 7 (1.1%) | <0.0001 |
| 10-year mortality | 8 (29.6%) | 105 (16.9%) | 0.115 |

suggest that VVB continues to be employed selectively in up to 38–42% of transplant centers across Europe and the United States [24, 25].

Among our most noteworthy findings, our results revealed that nearly one in five patients who underwent VVB procedures died in the immediate perioperative period. One alternative to VVB is the use of a temporary portocaval shunt (PCS). A PCS involves an end-to-side connection between the portal vein and infrahepatic IVC to reduce portal venous pressure, alleviate congestion in the splanchnic bed, and prevent intestinal edema. In our practice, the utilization of portocaval shunt (PCS) is not a standard procedure. In our institution, PCSs are employed in patients who do not have portal hypertension, for example, those with fulminant liver failure and no collateral blood vessels. This approach facilitates and optimizes hemodynamics during the anhepatic phase and helps prevent bowel edema. With VVB, we routinely place a cannula in the femoral vein and portal veins, obviating the necessity for a portocaval shunt. However, there is currently a contentious debate over the effectiveness of PCS in improving the outcomes of LT procedures. A recent systematic review with expert panel recommendations concluded that the routine use of PCS during LT procedures should be discouraged [6]. However, the expert panel did recognize a temporary PCS might be beneficial in certain rare cases and reiterated the need for multicenter, prospective randomized trials designed to delineate the immediate and short-term outcomes associated with its routine use. The results of several studies revealed that the implementation of PCSs is associated with a lower requirement for blood transfusion [7, 28–32] and reduced hospital lengths of stay [32].

Our case series provides important insight into the selective use of VVB during LT during the modern era at a major Australian transplant center. Among the unique features of our study, the indications for VVB at our center differ subtly from the broader range featured in previous reports. Within our cohort, elective VVB was performed nearly exclusively in patients with complex surgical anatomy; our series included only one case in which this procedure was performed in a patient with cardiac disease and concerns regarding hemodynamic stability.

Furthermore, at its inception, VVB was described as an alternative to complete caval clamping. As such, traditional indications for this procedure focused on clinical scenarios in which impaired venous return would be poorly tolerated, including pre-existing ventricular impairment, pulmonary hypertension, proven instability after test-clamping, and limited venous collateralization (i.e., fulminant liver failure and/or non-cirrhotic metabolic disease) [21, 33]. At our center, we predominantly employ methods that support caval preservation; thus, these indications are arguably less relevant.

Of note, our series included one patient with Budd-Chiari syndrome, which is historically described as a contraindication to VVB, given the potential increase in thromboembolic risk [2]. Shaw et al. [2] previously reported a case of fatal pulmonary embolism during the initiation of a bypass procedure in a patient with polycystic liver disease and massive hepatomegaly with IVC obstruction. They attributed the fatality to the contributions of a pre-existing thrombus and advocated for caution when performing VVB procedures in these patients. Interestingly, 40% of our case cohort (n = 12) included nine patients with polycystic liver disease and an additional three patients with massive hepatomegaly secondary to other causes. While we observed no intraoperative thrombotic complications during VVB, one patient experienced a postoperative pulmonary embolus on day 6 that did not result in hemodynamic compromise. The optimal approach to anticoagulation during VVB also remains to be determined. There is very little high-quality evidence supporting or refuting current systemic anticoagulation practices or the choice of different circuit types (i.e., heparin-bonded) [5].

Potentially life-threatening complications associated with vascular access needed for VVB are clearly documented in the literature [4, 34]. Two patients in our series sustained significant access-associated vascular injuries. One patient sustained a guidewire injury to the right heart which led to cardiac tamponade that required intraoperative evacuation. The patient ultimately recovered uneventfully. A second case involved an injury to the left subclavian artery that resulted in a massive hemothorax. A large-gauge venous "return" cannula was inserted into the left internal jugular vein. Although the left axillary vein can be used for venous access via a surgical cut-down on an elective basis, it was not considered in this case because of ongoing massive bleeding. The right internal jugular vein was also not suitable for cannulation as it was the access site used for the delivery of fluids and high-dose vasoconstrictor agents. In this case, VVB was attempted as a rescue technique during uncontrollable abdominal bleeding from collateral vessels. Line insertion was technically difficult due to concurrent hypovolemia and venous collapse. Ultimately, the patient experienced an asystolic arrest secondary to hemorrhagic shock and died intraoperatively. This case emphasizes the amplified risks of vascular complications while establishing rescue bypass from the left internal jugular vein, even while using ultrasound and transesophageal echocardiography to guide the cannulation and catheter placement. For these reasons, the right internal jugular vein is our institution's preferred access site for the VVB "return" cannulae. Compared to the left internal jugular vein, the right jugular vein runs more superficially and has a larger diameter [35] with an unhindered straight passage to the right atrium. Although transesophageal echocardiography is invaluable for guiding line insertion [36], as demonstrated in this case, its use does not completely preclude the development of vascular complications.

Clinical scenarios prompting the consideration of emergency bypass inherently involve co-existing physiologic insults, including hemorrhagic shock, prolonged vascular clamping, and organ hypoperfusion. Of the five in-hospital deaths included in our series, four were emergency rescue VVB cases. The issue of selection bias in VVB cohort studies has been acknowledged (38); this phenomenon may be further intensified when addressing the subset of patients who underwent bypass procedures as a rescue intervention. For example, results from previous studies revealed that patients who underwent VVB procedures had significantly

higher MELD scores compared to those who did not [13, 37]. By contrast, the VVB patients featured in our case series exhibit a relatively modest median MELD score of 20. The comparatively low MELD score reported for patients in our series reflects the large proportion of polycystic cases. However, in isolation, MELD scores may not fully characterize predicted surgical complexity, which was an overriding issue in our VVB cohort and a potential contributor to patient morbidity.

Overall, our complication rates were comparable to those reported in other VVB studies [21]. Apart from the major vascular access complications described above, our cohort also included one case of non-fatal air embolism, which presented as self-limiting hypoxia and transient right ventricular dysfunction. Although air emboli as a complication of VVB have been documented in previous studies [3], its overall incidence may be underestimated. Of note, we demonstrated a comparatively low rate of local complications that develop due to open vascular access. While 78% of systemic venous drainage lines at our center were achieved via femoral-saphenous cut-downs, nine patients (33%) underwent open axillary access for the placement of venous return cannulae. Nonetheless, we identified only one case of a wound-site seroma, which required no further management. Similarly, the median number of PRBCs transfused in patients in the VVB group was 7 units, which is consistent with the transfusion rates reported in other studies [20]. Finally, the use of VVB procedures was previously associated with the development of hypothermia [34]. Although 31% of our cohort experienced hypothermia (body temperature <35 ˚C at any point during the procedure), the median low temperature was a relatively robust 35.5 ˚C. This result may reflect the incorporation of a heat exchanger within the extracorporeal circuit.

Our study features several important strengths. The data collected for this study included a detailed exploration of baseline characteristics, intraoperative management, and postoperative outcomes. This enabled us to provide a comprehensive description of the pre- and perioperative state and evaluate how and for whom the practitioners at our center chose to implement VVB. Secondly, we have compared the preoperative characteristics and major outcomes of VVB patients to those who did not undergo this procedure.

The study also includes several limitations. Firstly, the findings reflect adult surgical practice at a single center and thus cannot be generalized to other institutions or pediatric LT recipients. Furthermore, the retrospective approach may have prevented us from capturing data from patients in whom VVB was deemed indicated but not initiated (e.g., due to inadequate vascular access or lack of an available perfusionist). It would be useful to identify these patients and explore the reasons why a planned VVB procedure was aborted as this would permit us to assess issues that affect the utilization of this resource. Finally, our case series is derived from a contemporary era with only a limited collection of long-term mortality data.

## Conclusion

In conclusion, VVBs were used in 3% of the LT procedures performed at a single center in Australia over a 14-year period. While elective indications predominantly reflected unique anatomical and surgical factors, emergency use was universally precipitated by massive intraoperative bleeding. The complications associated with VVB use were comparable to those reported in other studies. While inpatient mortality of patients who underwent VVB was significantly higher than in LT patients who did not undergo this procedure, we observed no significant differences in length of hospital stay as well as rates of acute kidney injury, primary graft nonfunction, and long-term graft failure. Patients who underwent a VVB procedure exhibited a higher rate of postoperative non-anastomotic biliary stricture. While VVB continues to play a vital role in LT in general, the global transition to selective VVB use underscores

the urgent need for collaborative multi-center studies designed to address outstanding questions related to patient selection and the safe implementation of this procedure.

## Supporting information

**S1 File. Intra and postoperative arterial blood gas data for veno-venous bypass patients.**
Data are presented as median (interquartile range) and [minimum maximum].
(DOCX)

## Author Contributions

**Conceptualization:** Laurence Weinberg, Dong-Kyu Lee, Marcos V. Perini.

**Data curation:** Riley Hazard, Jarryd Ludski, Hugh Slifirski, Patrick Nugraha, Daniel Do, Wendell Zhang, Robert Nicolae, Peter Kaldas.

**Formal analysis:** Laurence Weinberg, Dong-Kyu Lee.

**Investigation:** Laurence Weinberg, Marcos V. Perini.

**Methodology:** Laurence Weinberg, Marcos V. Perini.

**Project administration:** Laurence Weinberg.

**Validation:** Laurence Weinberg, Hugh Slifirski, Daniel Do, Wendell Zhang, Peter Kaldas, Marcos V. Perini.

**Visualization:** Laurence Weinberg, Rebecca Caragata, Riley Hazard, Dong-Kyu Lee, Michael A. Fink, Marcos V. Perini.

**Writing – original draft:** Laurence Weinberg, Rebecca Caragata, Dong-Kyu Lee, Peter Kaldas, Michael A. Fink, Marcos V. Perini.

**Writing – review & editing:** Laurence Weinberg, Rebecca Caragata, Riley Hazard, Jarryd Ludski, Dong-Kyu Lee, Hugh Slifirski, Patrick Nugraha, Daniel Do, Wendell Zhang, Robert Nicolae, Marcos V. Perini.

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
