## [Decision Letter · Decision Letter 0]

5 Dec 2023

PONE-D-23-26660Veno-venous bypass for liver transplant adult recipients: a single center observational case seriesPLOS ONE

Dear Dr. Weinberg,

Thank you for submitting your manuscript to PLOS ONE. After careful consideration, we feel that it has merit but does not fully meet PLOS ONE’s publication criteria as it currently stands. Therefore, we invite you to submit a revised version of the manuscript that addresses the points raised during the review process. Please submit your revised manuscript by Jan 19 2024 11:59PM. If you will need more time than this to complete your revisions, please reply to this message or contact the journal office at plosone@plos.org. Please include the following items when submitting your revised manuscript:A rebuttal letter that responds to each point raised by the academic editor and reviewer(s). You should upload this letter as a separate file labeled 'Response to Reviewers'.A marked-up copy of your manuscript that highlights changes made to the original version. You should upload this as a separate file labeled 'Revised Manuscript with Track Changes'.An unmarked version of your revised paper without tracked changes. You should upload this as a separate file labeled 'Manuscript'.

We look forward to receiving your revised manuscript.

Kind regards,

Academic Editor

PLOS ONE

Journal Requirements:

**Additional Editor Comments:**

Please revise.

Reviewers' comments:

Reviewer's Responses to Questions

**Comments to the Author**

1. Is the manuscript technically sound, and do the data support the conclusions?

Reviewer #1: Yes

Reviewer #2: Partly

2. Has the statistical analysis been performed appropriately and rigorously? 

Reviewer #1: Yes

Reviewer #2: Yes

3. Have the authors made all data underlying the findings in their manuscript fully available?

Reviewer #1: Yes

Reviewer #2: Yes

4. Is the manuscript presented in an intelligible fashion and written in standard English?

Reviewer #1: Yes

Reviewer #2: Yes

5. Review Comments to the Author

Reviewer #1: The paper presents a high quality of processing in terms of methodology, contents and results. As regards the introduction, it would be appropriate to increase the number of references and the length of the introduction, also introducing the background paragraph to provide a broader overview of the existing scientific literature also through a rapid revision of the same.

From a methodological point of view, I recommend providing the inclusion and exclusion criteria for recruited subjects more explicit because it would give further added value to this high-quality work.

The results are well represented both graphically and from a descriptive point of view, however, English should be improved.

The discussion is well structured, but I would suggest: 1) to increase the references because they are very few compared to the contents reported; 2) the conclusions must also be better specified by writing a paragraph entitled conclusions, making known what future research developments may be.

Reviewer #2: Dear authors,

Thak you for the possibility to review your work. I have however some important issues to consider regarding both methodology and positioning this paper in the current literature:

- Regarding only the VVB patients the series is very limited with only 29 cases over 14 years...? Therefore it would be of more interest to readers to compare (matched or complete series) between patients with and without VVB. This might also give more insight in the high rate of complications in the VVB series: are they worthwhile for this group of patients as they were sicker, more at risk etc.... At this time the results just show quite bad results without having an idea on the real patient population involved. After all your MELD score, at least for European standards (?) are quite low, so not even involving the sickest of the sickest..... Please consider a re-analysis of all your data over this time period, maybe also alluding on technical changes that happened over these 14 years...?

- My second important issue with this series on VVB and the discussion is that the authors hardly mention, nor discuss the use of the tempoarary portocaval shunt in these patients, sselectively or standardly used, as this is a technique that has hardly no complications, saves time, and has as good as all, or not more advantages for patients' hemodynamics.... At least a significant discussion point should be written to compare your technique with this option.

6. PLOS authors have the option to publish the peer review history of their article (what does this mean?). If published, this will include your full peer review and any attached files.

Reviewer #1: No

Reviewer #2: **Yes: **Frederik Berrevoet

---

## [Author Response · Author response to Decision Letter 0]

14 Apr 2024

Response to Reviewers

Robert Jeenchen Chen MD,MPH

Academic Editor

PLOS ONE

Dear Professor Chen

Re: PONE-D-23-26660: Venovenous bypass in adult liver transplant recipients: a single-center observational case series

Thank you for considering our revised manuscript at PLOS ONE. 

We have included the following items in our revised manuscript:

• A rebuttal letter that responds to each point raised by the academic editor and reviewer(s). We have uploaded this as a separate file with the label 'Response to Reviewers'.

• A marked-up copy of our manuscript highlights changes made to the original version. All changes are marked in red. This is uploaded as a separate file labeled 'Revised Manuscript with Track Changes'.

• An unmarked version of your revised paper without tracked changes. This has been uploaded as a separate file labeled 'Manuscript'.

Reviewer ONE 

We thank Reviewer One for their expert insights and time spent reviewing our manuscript. We have provided a detailed comment to each of the reviewer’s questions, as outlined below. 

Reviewer 1: The paper presents a high quality of processing in terms of methodology, contents and results.

Authors’ response: Thank you for this positive comment. We hope our revised submission addresses all the important questions asked, and we hope that this case series makes a significant contribution to the existing literature.

Question 1: As regards the introduction, it would be appropriate to increase the number of references and the length of the introduction, also introducing the background paragraph to provide a broader overview of the existing scientific literature also through a rapid revision of the same.

Author response: Thank you for this excellent suggestion. Accordingly, we have made substantial changes to the introduction section, citing twelve additional references. We have rewritten the introduction section to provide a more comprehensive overview of the existing literature, as advised.

In the revised manuscript, we have added a separate paragraph that provides a broad overview of the use of venovenous bypass in liver transplant recipients. 

We now state “Many variations in VVB set-up and technique as well as numerous approaches to vascular access have been developed. (3,4)

The performance of caval anastomoses with or without VVB continues to be a subject of debate. A rigorous Cochrane analysis examined the advantages and disadvantages of this procedure and reported that there was insufficient evidence available to support or dismiss the use of VVB in LT. (5) As part of this analysis, the authors evaluated the evidence that directly compared LT procedures performed with or without VVB. Of note, no significant differences in renal failure or blood transfusion requirements were identified. However, all trials identified featured a high risk of bias, and none reported on patient or graft survival.

More recently, an expert panel at a consensus conference on Enhanced Recovery for Liver Transplantation evaluated the routine application of VVB during LT and identified length of hospital stay, duration of surgery, units of packed red blood cells (PRBCs) transfused, early complication rates and mortality, and renal dysfunction as critical significant outcomes for patients undergoing VVB. (6) A comprehensive re-evaluation of the current literature and assessments focused on VVB during LT revealed that the quality of evidence for all outcomes of significance was extremely poor. The authors reported that most of the studies that compared outcomes of LT procedures involving VVB to those with no VVB reported no significant differences in duration of surgery (7-13), units of PRBCs transfused (7-14) early postoperative mortality (9,12,14), postoperative acute kidney injury (7-9,11-13,15,16), early complication rates (10,12,16), or length of hospital stay. (8,11,12) The expert panel advised against the routine implementation of VVB, but acknowledged that there are specific circumstances in which its use may be justified.”

Question 2: From a methodological point of view, I recommend providing the inclusion and exclusion criteria for recruited subjects more explicit because it would give further added value to this high-quality work.

Authors’ response: Thank you for this excellent comment. This has been revised in the Methods section. We now state “The adult recipient program at this LT unit uses only deceased donor grafts. Inclusion criteria included adult patients (i.e., those ≥18 years of age) who received VVB as part of their LT procedure for any indication, including redo LTs, and patients who received elective (planned) or emergent (rescue) VVB. Exclusion criteria included paediatric LT recipients (i.e., those <18 years of age).”

Question 3: The results are well represented both graphically and from a descriptive point of view, however, English should be improved.

Authors’ response: Thank you for this comment. The manuscript has undergone comprehensive editing, with corrections to grammar, formatting and syntax. A professional editor has also reviewed the document.

Question 4: The discussion is well structured, but I would suggest: 1) to increase the references because they are very few compared to the contents reported; 2) the conclusions must also be better specified by writing a paragraph entitled conclusions, making known what future research developments may be.

Authors’ response: Thank you for this comment. Accordingly, we have made substantial changes to the introductions section, citing an additional twelve references. As advised, the introduction section has been rewritten to include a broader overview of the existing literature, as outlined above.

Additionally, we have extended the concluding paragraph, which now reads “In conclusion, VVBs were used in 3% of the LT procedures performed at a single center in Australia over a 14-year period. While elective indications predominantly reflected unique anatomical and surgical factors, emergency use was universally precipitated by massive intraoperative bleeding. The complications associated with VVB use were comparable to those reported in other studies. While inpatient mortality of patients who underwent VVB was significantly higher than in LT patients who did not undergo this procedure, we observed no significant differences in length of hospital stay as well as rates of acute kidney injury, primary graft nonfunction, and long-term graft failure. Patients who underwent a VVB procedure exhibited a higher rate of postoperative bile duct stricture. While VVB continues to play a vital role in LT in general, the global transition to selective VVB use underscores the urgent need for collaborative multi-center studies designed to address outstanding questions related to patient selection and the safe implementation of this procedure”

Reviewer Two

We thank Reviewer Two for their expert insights and time spent reviewing our manuscript. We have provided a detailed comment to each of the reviewer’s questions, as outlined below.

Reviewer 2: Thank you for the possibility to review your work. I have however some important issues to consider regarding both methodology and positioning this paper in the current literature:

Authors’ response: Thank you for this comment. We thank Professor Berrevoet for taking the time to review our manuscript. We are very grateful that such a distinguished clinician and academic has provided us with excellent insights. 

Question 1: Regarding only the VVB patients the series is very limited with only 29 cases over 14 years. Therefore, it would be of more interest to readers to compare (matched or complete series) between patients with and without VVB. This might also give more insight in the high rate of complications in the VVB series: are they worthwhile for this group of patients as they were sicker, more at risk etc.... At this time the results just show quite bad results without having an idea on the real patient population involved. 

Authors’ response: Thank you for this comment. We have now included data on the patients who did not receive VVB and updated the manuscript accordingly. 

We now state “Of the 900 LTs performed at this center during the aforementioned 14-year period, 27 (3%) included a VVB procedure. VVB was performed electively in 16 of these 27 patients (59.3%) and as a rescue technique to control massive bleeding in the other 11 (40.1%). The median (interquartile range [IQR]) age of those who underwent VVB procedures was 48 (39–55) years; the median age was 56 (47–62) years in the non-VVB group (p<0.0001). The median Model for End-Stage Liver Disease (MELD) scores were similar between the two patient groups. Twenty-six VVB (96.3%) and 603 non-VVB (96.9%) patients required intraoperative blood transfusions. The median (IQR) number of units of packed red blood cells transfused was 7 (4.8–12.5) units in the VVB group compared to 3.0 units (1.0–6.0) in the non-VVB group (p<0.0001). Inpatient mortality was 18.5% and 1.1% for the VVB and non-VVB groups, respectively (p<0.0001). There were no significant differences in length of hospital stay or incidence of acute kidney injury, primary graft dysfunction, or long-term graft failure between the two groups. Patients in the VVB group experienced a higher rate of postoperative bile duct stricture compared to patients in the non-VVB group (33% and 7.9%, respectively; p=0.0003). 

We have also updated Table 1 to reflect the differences between patients receiving VVB versus those who did not. 

Further, we have added to Table 3 a comparison of fluids, whole blood, and blood products used by patients receiving VVB versus those who did not. We included 622 patients with complete medical records who underwent transplants during the same period. Data are presented as numbers (proportions), medians (interquartile ranges), and [minimum and maximum].

Table 3. Fluids, whole blood, and blood products provided. Only patients with complete medical records were included. Data are presented as number (proportion), median (interquartile ranges), and [minimum and maximum].

Question 2: Your MELD score, at least for European standards are quite low, so not even involving the sickest of the sickest. Please consider a re-analysis of all your data over this time period, maybe also alluding on technical changes that happened over these 14 years.

Authors’ response: Thank you for this important comment. As discussed in the manuscript, results from previous studies revealed that patients who underwent VVB procedures had significantly higher MELD scores compared to those who did not. By contrast, the VVB patients featured in our case series exhibit a relatively modest median MELD score of 20. The comparatively low MELD score reported for patients in our series reflects the large proportion of polycystic cases. However, in isolation, MELD scores may not fully characterize predicted surgical complexity, which was an overriding issue in our VVB cohort and a potential contributor to patient morbidity. This has been included in the manuscript. There were no significant technical changes over the time period, and the core surgical, anesthesia and intensive care clinicians caring for these patients over the 14-year period were unchanged. team providing. The use of thromboelastography to guide the use of blood products has been used in our institution for liver transplants since 2000.

We have also analysed the patients’ MELD scores over time. As presented in the figure, there were no clinically meaningful or statistically significant changes in MELD scores over the study period. The resubmitted manuscript also incorporates this data.

Question 3: An important issue with this series on VVB and the discussion is that the authors hardly mention, nor discuss the use of the temporary portocaval shunt in these patients, selectively or standardly used, as this is a technique that has hardly no complications, saves time, and has as good as all, or not more advantages for patients' hemodynamics. At least a significant discussion point should be written to compare your technique with this option.

Authors’ response: Thank you for this very insightful comment. As suggested, we have now included an additional section in our discussion outlining our use of portocaval shunts. When using the VVB at our institution, we always place a Y-connection so that the portal and caval blood flow will join and run back to the venous inflow cannula in the internal jugular or axillary vein. When using the VVB, there is therefore no need for a temporary portocaval shunt (i.e., the portal vein has already been used). We consider selectively using the portocaval shunt in patients without portal hypertension (i.e., fulminant liver failure) and no collaterals in order to allow some flow back through the natural shunts. The reason we use portocaval shunts in such selective cases is to avoid bowel oedema, which in turn reduces space in the abdominal cavity, facilitating abdominal closure, although we are cognizant that there is no strong evidence to support this statement. Further, the evidence that the use of portocaval shunt improves outcomes is very limited. This has been discussed in our resubmission manuscript. A recent panel of experts revised the technical aspects of liver surgery and were against the routine use of PCS. We have also updated our references to include this paper. 

In our revised manuscript we now state “Among our most noteworthy findings, our results revealed that nearly one in five patients who underwent VVB procedures died in the immediate perioperative period. One alternative to VVB is the use of a temporary portocaval shunt (PCS). A PCS involves an end-to-side connection between the portal vein and infrahepatic IVC to reduce portal venous pressure, alleviate congestion in the splanchnic bed, and prevent intestinal edema. In our practice, the utilization of portocaval shunt (PCS) is not a standard procedure. In our institution, PCSs are employed in patients who do not have portal hypertension, for example, those with fulminant liver failure and no collateral blood vessels. This approach facilitates the restoration of blood flow through natural shunts, optimizes hemodynamics during the anhepatic phase, and prevents bowel edema. With VVB, we routinely access the inferior vena cava via a saphenous or femoral approach, as well as the portal vein, obviating the necessity for a portocaval shunt. However, there is currently a contentious debate over the effectiveness of PCS in improving the outcomes of LT procedures. A recent systematic review with expert panel recommendations concluded that the routine use of PCS during LT procedures should be discouraged. (6) However, the expert panel did recognize a temporary PCS might be beneficial in certain rare cases and reiterated the need for multicenter, prospective randomized trials designed to delineate the immediate and short-term outcomes associated with its routine use. The results of several studies revealed that the implementation of PCSs is associated with a lower requirement for blood transfusion (7, 28-32) and reduced hospital lengths of stay. (32)

Once again, we would like the expert reviewers for taking the time to review and consider our manuscript for publication in PLOS ONE. 

Professor Laurence Weinberg

BSc, MBBCh, MRCP, DPCritCareEcho, FANZCA, MD, PhD

Director, Department of Anaesthesia, Austin Hospital

Professor, Department of Critical Care, University of Melbourne

---

## [Decision Letter · Decision Letter 1]

30 Apr 2024

Venovenous bypass in adult liver transplant recipients: a single-center observational case series

PONE-D-23-26660R1

Dear Dr. Weinberg,

We’re pleased to inform you that your manuscript has been judged scientifically suitable for publication and will be formally accepted for publication once it meets all outstanding technical requirements.

Kind regards,

Academic Editor

PLOS ONE

Additional Editor Comments (optional):

Reviewers' comments:

Reviewer's Responses to Questions

**Comments to the Author**

1. If the authors have adequately addressed your comments raised in a previous round of review and you feel that this manuscript is now acceptable for publication, you may indicate that here to bypass the “Comments to the Author” section, enter your conflict of interest statement in the “Confidential to Editor” section, and submit your "Accept" recommendation.

Reviewer #1: All comments have been addressed

Reviewer #2: All comments have been addressed

2. Is the manuscript technically sound, and do the data support the conclusions?

Reviewer #1: Yes

Reviewer #2: Yes

3. Has the statistical analysis been performed appropriately and rigorously? 

Reviewer #1: Yes

Reviewer #2: Yes

4. Have the authors made all data underlying the findings in their manuscript fully available?

Reviewer #1: Yes

Reviewer #2: Yes

5. Is the manuscript presented in an intelligible fashion and written in standard English?

Reviewer #1: Yes

Reviewer #2: Yes

6. Review Comments to the Author

Reviewer #1: The paper has a good structure and the revisions are well written. The methods are in line with the current literature. The paper can be published.

Reviewer #2: Personally I believe the quality of the manuscript improved significantly after the suggestions of both reviewers. As a personal note, I still can't agree with the comments regarding the portocaval shunt, as it is much easier to perform than a VV bypass, with overall comparable positive effects on hemodynamics, blood transfusion, need for u perfusionist etc.... but again, just a personal opinion, not a proven fact.

Congratulations to the authors for their well performed analysis.

7. PLOS authors have the option to publish the peer review history of their article (what does this mean?). If published, this will include your full peer review and any attached files.

Reviewer #1: No

Reviewer #2: **Yes: **Frederik Berrevoet
